# Prevalence and factors associated with early resumption of sexual intercourse among postpartum women: Systematic review and meta-analysis

**Kelemu Abebe Gelaw** [1]*, **Yibeletal Assefa Atalay**[2], **Adisu Yeshambel**[1], **Getachew Asmare Adella**[3], **Belete Gelaw Walle**[4], **Liknaw Bewket Zeleke**[5,6], **Natnael Atnafu Gebeyehu**[1]

1 School of Midwifery, College of Health Science and Medicine, Wolaita Sodo University, Wolaita Sodo, Ethiopia, 2 School of Public Health, College of Health Science and Medicine, Wolaita Sodo University, Wolaita Sodo, Ethiopia, 3 Department of Reproductive Health, College of Health Science and Medicine, Wolaita Sodo University, Wolaita Sodo, Ethiopia, 4 School of Nursing, College of Health Science and Medicine, Wolaita Sodo University, Wolaita Sodo, Ethiopia, 5 Health Science College, Debre Markos University, Debre Markos, Ethiopia, 6 School of Women's and Children's Health, University of New South Wales Sydney, Kensington, Australia

* kelemuabebe2014@gmail.com, kelemu.abebe@wsu.edu.et

**Data Availability Statement:** All relevant data are within the paper and its supporting Information files.

## Abstract

### Introduction

Postpartum sexual health is indicated by a resumption of sexual activity as well as arousal, desire, orgasm, and sexual satisfaction. The issue of resuming sexual intercourse after childbirth has received limited attention because healthcare professionals rarely provide adequate care to postnatal women. The present study aimed to ascertain the overall prevalence of early resumption of sexual intercourse among most women.

### Methods

Searches were conducted in PubMed, Web of Science, Science Direct, Google Scholar, African Journals Online, and the Cochrane Library. Data were extracted using Microsoft Excel, and STATA version 14 was used for analysis. Publication bias was checked by funnel plot, Egger, and Begg regression tests. A p-value of 0.05 was regarded to indicate potential publication bias. Using $I^2$ statistics, the heterogeneity of the studies was evaluated. By country, a subgroup analysis was conducted. A sensitivity analysis was carried out to determine the effect of each study's findings on the overall estimate. The random effects model was used to assess the overall effect of the study and then measured using prevalence rates and odds ratio with 95% CI.

### Results

Twenty-one studies with 4,482 postpartum women participants were included in the study. The pooled prevalence of early resumption sexual intercourse among post-partum women was 57.26% (95% CI 50.14, 64.39) with significant heterogeneity between studies ($I^2$ = 99.2%; P-value ≤ 0.000) observed. Current contraceptive use (AOR = 1.48, 95%CI = 1.03,

**Funding:** The authors received no specific funding for this work.

**Competing interests:** The authors have declared that no competing interests exist.

6.21), primipara (AOR = 2.88, 95%CI = 1.41, 5.89), and no history of severe genital injury on the last delivery (AOR = 2.27, 95%CI = 1.05, 4.93) were significantly associated with early resumption of sexual intercourse.

## Conclusion

This study found that more than half of women resumed sexual intercourse early after giving birth. This suggests that a significant number of women may be at higher risk of unwanted pregnancies, short birth intervals, and postpartum sepsis. Thus, stakeholders should improve the integration of postpartum sexual education with maternal health services to reduce the resumption of postpartum sexual intercourse.

## Introduction

Sexuality evolves throughout a person's life and is essential to being human [1]. It is a key idea in human sexual health [2]. Sexual health is a key component of the overall health and quality of life of both men and women [3]. It has at least three connected purposes: communication, pleasure, and reproduction. Pregnancy, childbirth, and the postpartum period are crucial times for women's sexual health [4]. Women's bodies experience physiological as well as anatomical changes after giving birth which have an impact on their lives, including sexual activity, which is essential to human existence [5]. Postpartum sexual health is indicated by resuming sexual activity as well as by arousal, desire, orgasm, and sexual satisfaction [6]. World Health Organization (WHO) advises that postpartum sexual health is one of the most crucial issues that need to be addressed during the postpartum period [7].

The desire to engage in sexual activity declines during pregnancy but returns to normal, usually six weeks after delivery [8]. Resuming postpartum sexual activity should be done at least six weeks after giving birth [9]. However, the decision to resume sexual activity after delivery differs from woman to woman and is influenced by several factors such as the quantity of bleeding, mode of delivery, culture, maternal mental health, infant health, the relationship with one's partner, and the mother's general health [10].

Early resumption of sexual intercourse among postpartum women may cause maternal health problems: unfavorable birth outcomes, wound infection, painful intercourse, vaginal dryness, and inability to achieve orgasm [11]. In addition, early resuming sexual intercourse without utilizing contraception after giving birth can lead to an unplanned pregnancy and short birth interval. Due to this, there may be a high risk of death for infants under the age of one year [12]. Sociocultural norms and beliefs, education, the place of the birth, the mother's breastfeeding status, and mode of delivery, low parity, use of contraceptives, and residence are some of the factors that have been found to affect the early resumption of sexual activity after delivery [13].

The previous study has provided evidence that the use of hands-free perineal control techniques during the second stage of labor may represent a promising delivery approach to maintain perineal integrity. In addition, these techniques can have a positive effect on the resumption of sexual activity in the postpartum period, as they include several factors that can facilitate this process [14]. Numerous cohort studies have shown that women who undergo spontaneous vaginal delivery with intact perineum have a higher likelihood of having vaginal intercourse again within six to eight weeks after birth compared to women who undergo an episiotomy assisted vaginal birth etc. undergo a Cesarean Section [15–17]. Even though the

WHO advises that all women should be evaluated 2–6 weeks after childbirth, the problem of early resumption of sexual activity during postpartum has received little attention from researchers, policymakers, and healthcare providers [18]. Therefore, it is important to examine the various aspects of early resumption of sexual relationships in the postpartum period to understand the direct and indirect effects of this problem and highlight the importance of studying these issues at a global level [19]. Furthermore, the prevalence of early resumption of sexual intercourse varied in the previous studies, with rates ranging from 20.2% in Ethiopia [42] to 90.2% in the USA [24]. Given these differences, there is no systematic review or meta-analysis conducted on the prevalence and contributing variables associated with early postpartum sexual resumption worldwide.

Additionally, the results of this study might help in developing new approaches to postpartum sexuality education. Hence, this study aimed to assess the prevalence and factors associated with early resuming postpartum sexual intercourse at the global level.

## Methods

### Search strategy and information sources

An extensive data search was performed on International online databases (PubMed, Web of Science, Science Direct, Google Scholar, Cochrane Library, and African Journals Online (AJOL) databases used to get the research articles. We also retrieved gray literature from Addis Ababa and Bitesema University's online research institutional repository. Searching strategies were established by using Boolean operators ("OR" or "AND") and the following key terms: early sexual intercourse, associated factors, and post-partum. The search strategies for Google Scholar were: "early sexual intercourse" and ("associated factors") and "post-partum". We also retrieved gray literature from Addis Ababa and Busitema University's online research institutional repository. During the search process, the following phrases and keywords were used: "prevalence," "incidence," "magnitude," "early resumption," "sexual intercourse," "determinant," "factor," "predictor," and "postpartum period" The last date of search that all databases were checked from February 15/2023 to February 27/2023.

### Reporting

We used PICO questions that had been modified to follow the PEO (Population, Exposure, and Outcome) style for the explicit presentation of our review question and the explicit clarification of the inclusion and exclusion criteria. We reported according to Preferred Reporting Items for Systematic Reviews and Meta-Analyses (PRISMA) criteria for conducting the systematic review [20] (**S1 Table**). The authors of this systematic review and meta-analysis work were registered with Prospero at CRD42023427034.

### PEO guide

**P: Population (Patients).**
✓ Postpartum women who had early resumption of sexual intercourse.
**E: Exposure.**
✓ Factors that affect for early resumption of sexual intercourse among postpartum women.
**O: Outcome.**
✓ Prevalence of early resumption of sexual intercourse among postpartum women.
✓ Factors associated with early resumption of sexual intercourse among postpartum women.
**Study population.**
✓ Postpartum women.

### Eligibility criteria

Studies that included full-text articles, English-language articles, both published and unpublished articles, cross-sectional studies, case-control studies, and cohort studies were included in this study. Duplicate sources, interventional studies, case reports, systematic reviews, qualitative articles, case series, conference abstracts, letters to the editor, and any articles that were not fully accessible after exchanging at least two emails with the lead author were all excluded. A COCOPOP (Condition, Context, and Population) paradigm was utilized to determine the suitability of the included studies for this investigation. Postpartum women who had early resumption of sexual intercourse made up the study's population (POP), while the prevalence of early resumption of sexual intercourse served as the condition and global served as the setting.

### Outcome measurements

This review and meta-analysis had two main outcomes. The primary outcome was the global prevalence of early resumption of sexual intercourse among postpartum women. The second outcome was factors associated with the early resumption of sexual intercourse among postpartum women. We used the following variables for factors: contraceptive use, multiparous, maternal educational status, sexual intercourse during pregnancy, history of genital injury on the last delivery, mode of delivery, breastfeeding status, the onset of menstruation, husband educational status, and timing resumption of intercourse if they were listed as a factor in at least two studies. Data were taken in the form of two two-by-two tables from the primary studies for each factor to calculate the odds ratio.

### Study selection and data extraction

Retrieved articles were exported to the reference manager software; endnote software was used to remove duplicate studies. Three independent reviewers screened the title and abstract (KA, YA, and NA). The disagreement was handled based on one established article selection criteria. Data were extracted using a standardized data extraction format prepared in Microsoft Excel by five independent authors (KA, AY, BG, GA, and LB). Any difference during extraction was solved through discussion. The name of the first author, study area and country, the study design, year of publication, study design, study setting, sample size, and prevalence of early resumption of sexual intercourse among postpartum women were collected.

### Risk of bias (Quality assessment)

The scientific validity and quality of each study were assessed using the Joanna Briggs Institute (JBI) quality assessment approach, which is designed for cross-sectional, case-control, and cohort studies. Each author assessed each study separately using the above assessment method. Analyzes for cross-sectional studies were performed using assessment results that met (4 of 8) a 50% low-risk quality assessment requirement. For case-control and cohort studies, assessments (5 of 10) and (5 of 11) each met a low-risk quality assessment requirement of 50% (S2 Table). Two independent authors (KA and YA) assessed the quality of the study. Any disagreements raised during the bias assessment were resolved through a discussion led by the third author (NAG).

### Publication bias, heterogeneity, and statistical analysis

Data were extracted using Microsoft Excel and analyzed using STATA version 14 statistical software. The presence of significant between-study heterogeneity was assessed using

Cochrane Q and $I^2$ statistics. The presence of heterogeneity was illustrated by a forest plot. We utilized a random-effect model for analysis to estimate the pooled effect because we found a high level of heterogeneity. Analysis of the subgroups was done by study setting, study design, and country. A sensitivity analysis was carried out to determine the effect of one study's findings on the overall estimate. To detect the presence of considerable heterogeneity, meta-regression was computed based on publication year and country. Publication bias was checked by funnel plot and Egger's regression tests. At a p-value of less than 0.05, publication bias was considered to be statistically significant. We used adjusted odds ratio estimates with confidence intervals (CI) as a measure of association. The random effect model was used to assess the overall effect of early resumption of sexual intercourse, which was then measured by the prevalence rates and odds ratio with 95% CI. The result was presented in the form of text, tables, and figures.

## Result

### Selection of included studies

A PubMed, Web of Science, Science Direct, Google Scholar, Cochrane Library, and African Journals Online AJOL resulted in a total of 8,460 research articles. Among these studies, 6,850 duplicate studies were removed, and 1,510 studies were excluded after reviewing their titles and abstracts. At the eligibility evaluation phase, out of the remaining 100 studies, 79 articles were removed after examining their full text, and similar by considering the inclusion and exclusion criteria. Lastly, 21 studies [21–41] and 4,482 participants were included in the analysis(Fig 1).

### Description of included studies

Table 1 displays the characteristics of all included studies. The author's name, publication year, study setting, study design, sample size, country, and the percentage of early resumption of sexual intercourse In terms of country-wise distribution, the included 21 studies were comprised of 2 from the United States of America (23 and 24), 5 from Ethiopia (39,40,41,42, and 43), 4 from Uganda(35,36,37 and 38), 3 in Nigeria (32,33, and 34), 2 from China (25 and 26), 2 from India (30 and 31), and the rest 3 were from Spain (27), Norway (28), and Iran (29). None of the studies were excluded based on the quality assessment criteria (Table 1).

### Meta-analysis

**Prevalence of early resumption of sexual intercourse among postpartum women.**
From these reviewed studies, the prevalence of early resumption of sexual intercourse among post-partum women ranged from 20.2 to 90.2% [19, 37]. The global pooled prevalence of early resumption of sexual intercourse among post-partum women was 57.26% (95% CI 50.14, 64.39). The random-effect model was used to analyze the pooled prevalence; however a high and significant heterogeneity among the included studies ($I^2$ = 99.2%; P-value ≤ 0.000) was observed (Fig 2).

### Subgroup analysis

After confirming the presence of heterogeneity among the studies, subgroup analysis was done based on the study setting, country, and study design to identify the source of heterogeneity. Nevertheless, there was still proof of study heterogeneity. In the sub-group analysis, the United States of America had the highest prevalence of early resumption of sexual intercourse among postpartum women(75.670%; 95% CI: 46.77; 104.58) while China had the lowest prevalence the pooled prevalence of early resumption of sexual intercourse among post-partum (45.98;26.48, 65.48) (**Table 2**).

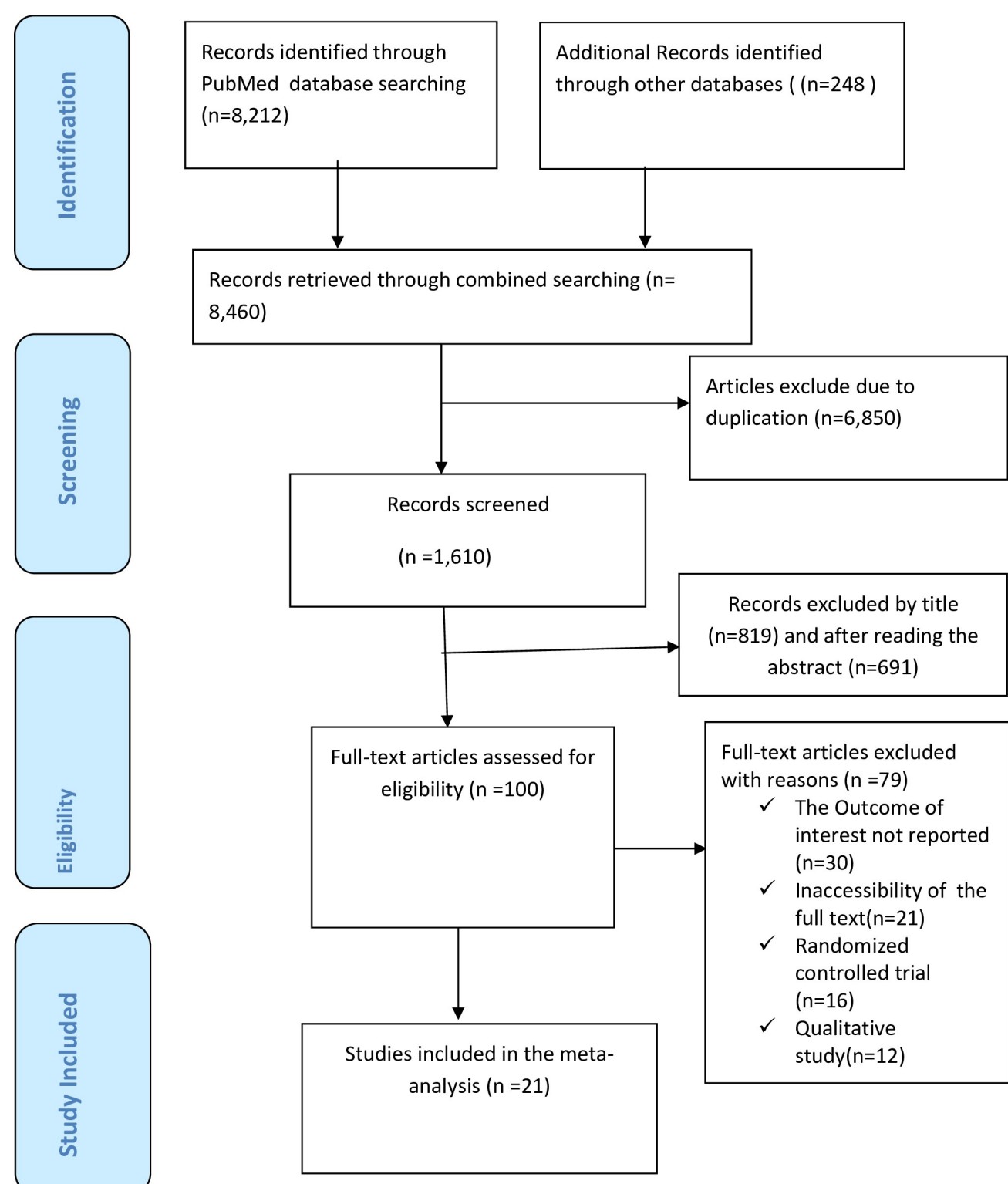

**Fig 1. PRISMA flow chart displays studies used for systematic review and meta-analysis of prevalence and factors associated with early resumption of sexual intercourse among postpartum women.**

**Table 1. Descriptions of the studies used in the systematic review and meta-analysis for prevalence and factors associated with early resumption of sexual intercourse among postpartum women.**

| First Author/year | Study Setting | Country | Study design | Prevalence((95%CI) | Sample Size | Study Quality |
|---|---|---|---|---|---|---|
| Yee et al./2013 [21] | health facility | USA | Cohort | 60.7 | 160 | Low risk |
| Brubaker L et al./2008 [22] | health facility | USA | Cohort | 90.2 | 509 | Low risk |
| Xiaorong Fan /2021 [23] | health facility | China | cross-sectional | 55.9 | 15, 834 | Low risk |
| Caixia Zhuang et.al/2019 [24] | Community | China | cross-sectional | 36 | 550 | Low risk |
| Sònia Anglès et.al/2019 [25] | health facility | Spain | case-control | 73 | 318 | Low risk |
| Kathrine Fodstad et.al/2016 [26] | health facility | Norway | case-control | 51.4 | 2848 2846 | Low risk |
| Prakash P 2021 [27] | health facility | India | cross-sectional | 41 | 3,112 | Low risk |
| Gyan P et.al/2021 [28] | Community | India | cross-sectional | 65 | 1564 | Low risk |
| Fatemeh D/2014 [29] | health facility | Iran | cross-sectional | 68 | 150 | Low risk |
| Olugbenga Bello et.al/2017 [30] | health facility | Nigeria | cross-sectional | 45.2 | 460 | Low risk |
| Anzaku AS eta.al/2014 [31] | health facility | Nigeria | cross-sectional | 67.6 | 340 | Low risk |
| Kola M/2014 [32] | health facility | Nigeria | cross-sectional | 40 | 257 | Low risk |
| Alice C et. al/2015 [33] | health facility | Uganda | cross-sectional | 21.6 | 374 | Low risk |
| Rose N et.al/2021 [34] | health facility | Uganda | Cohort | 88.2 | 507 | Low risk |
| MADENJE M et.al/2019 [35] | Community | Uganda | cross-sectional | 25 | 622 | Low risk |
| Emmanuel O et.al/2003 [36] | health facility | Uganda | cross-sectional | 66.4 | 216 | Low risk |
| Tariku B et.al/2021 [37] | health facility | Ethiopia | cross-sectional | 53.9 | 330 | Low risk |
| Dejene E/2022 [38] | health facility | Ethiopia | cross-sectional | 31.6 | 424 | Low risk |
| Melaku H/2022 [39] | Community | Ethiopia | cross-sectional | 60.4 | 6447 | Low risk |
| Ebisa Turi et.al/2022 [40] | health facility | Ethiopia | cross-sectional | 20.2 | 528 | Low risk |
| Frewoini T/2014 [41] | health facility | Ethiopia | cross-sectional | 78.3 | 424 | Low risk |

## Sensitivity analysis

The influence of each study on the overall prevalence of early resumption of sexual intercourse among postpartum women was examined using a leave-one-out sensitivity analysis, which involved removing one study at a time. The outcome showed that the omitted study had no significant effect on postpartum women's early resumption of sexual activity (Table 3).

## Meta-regression

Meta-regression was done using publication year and country to test for underlying the source of heterogeneity. There was no evidence to support publication year, and country were the causes of the heterogeneity (p-value = 0.723) and (p = 0.16), respectively (Table 4).

## Publication bias

The results of the funnel plot demonstrate that there was no publication bias present in any of the studies, which was an asymmetric distribution. In addition, both the funnel plot and Eggers regression test were used to determine publication bias in included studies. However, no evidence of publication bias was found by the Eggers and Begg regression test with p-values of 0.422 and 0.386, respectively (Fig 3).

## Factors associated with early resumption of sexual intercourse among postpartum women

The association between the prevalence of early resumption of sexual intercourse among postpartum women and the use of contraception was evaluated by using four studies [33, 34, 37,

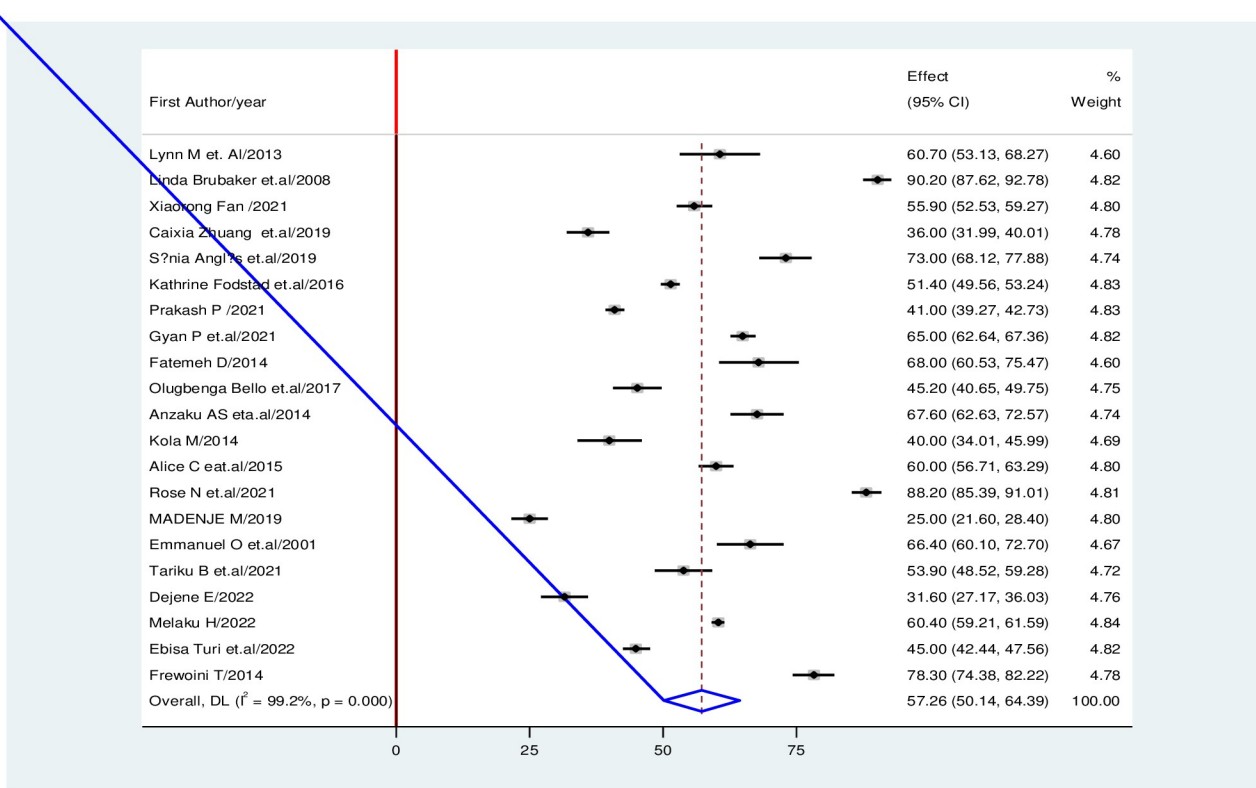

**Fig 2. Forest plot of the prevalence with corresponding 95% CIs of the twenty-one studies on prevalence and factors associated with early resumption of sexual intercourse among postpartum women.**

**Table 2. The pooled prevalence of early resumption of sexual intercourse among postpartum women, 95% CI, and heterogeneity estimate with a *p*-value for subgroup analysis.**

| Country | Random effects(95%CI) | Test of heterogeneity I² |
|---|---|---|
| USA | 75.67(46.77, 104.58) | 98.1 |
| China | 45.98(26.48, 65.48) | 98.2 |
| Spain | 73.00 (68.120, 77.88) | ---- |
| Norway | 51.40 (49.57, 53.24) | ---- |
| India | 52.98 (29.47, 76.51) | 99.6 |
| Iran | 68.00 (60.54,75.47) | ------ |
| Nigeria | 50.97 (34.50, 67.45) | 99.8 |
| Uganda | 59.89 (30.65, 89.14) | 99.6 |
| Ethiopia | 53.86 (41.70, 66.04) | 99.9 |
| Overall | 57.26 (50.14,64.39) | 99.2 |
| **Study setting** | | |
| Health Facility | 59.78 (50.87,68.75) | 99.2 |
| Community | 46.67 (30.67,62.68) | 99.4 |
| Overall | 57.26 (50.14,64.39) | 99.2 |
| **Study design** | | |
| Cohort | 80.60 (70.04,91.16) | 96.2 |
| Cross-sectional | 52.40(45.72,59.06) | 98.7 |
| Case-control | 62.08(40.91, 83.24) | 98.5 |
| Overall | 57.26 (50.14, 64.39) | 99.2 |

**Table 3. Sensitivity analysis for prevalence and factors associated with early resumption of sexual intercourse among postpartum women.**

| Study omitted | Estimate | (95%CI) |
|---|---|---|
| Lynn M et. Al/2013 | 57.1 | 49.77, 64.42 |
| Linda Brubaker et.al/2008 | 55.59 | 49.11, 62.06 |
| Xiaorong Fan /2021| | 57.33 | 49.87, 64.78 |
| Caixia Zhuang et.al/2019 | 58.32 | 51.08-,65.57 |
| Sonia Angles et.al/2019 | 56.48 | 49.16,63.7 |
| Kathrine Fodstad et.al/2016 | 57.56 | 57.56,65.31 |
| Prakash P /2021 | 58.08 | 50.8,65.32 |
| Gyan P et.al/2021 | 56.87 | 49.32,64.42 |
| Fatemeh D/2014 | 56.74 | 49.42–64.06 |
| Olugbenga Bello et.al/2017 | 57.86 | 50.51,65.21 |
| Anzaku AS eta.al/2014 | 56.74 | 49.40,64.09 |
| Kola M/2014 | 58.11 | 50.81,65.41 |
| Alice C et al./2015 | 57.12 | 50.81,65.41 |
| Rose N et al./2021 | 55.69 | 48.95,62.43 |
| MADENJE M/2019 | 55.69 | 48.95,62.43 |
| Emmanuel O et al./2001 | 56.81 | 49.48,64.14 |
| Tariku B et al./2021 | 57.43 | 50.07,64.79 |
| Dejene E et al./2022 | 58.54 | 51.35,65.73 |
| Melaku H/2022 | 57.1 | 57.1,65.41 |
| Ebisa Turi et al./2022 | 57.26 | 50.13,64.39 |
| Frewoini T/2014 | 56.20 | 48.96, 63.45 |
| Overall | 57.26 | 50.13,64.39 |

39]. The result revealed that the pooled effect of the use of contraception was significantly associated with the early resumption of sexual intercourse among post-partum women. Women who used contraceptives were 1.48 times more likely to have an early resumption of sexual intercourse during the postpartum period than those who didn't use (AOR = 1.48, 95% CI = 1.03, 6.2.12). Heterogeneity was not detected across the studies (I-squared = 0.0%, p = 0.42 (Fig 4).

The association between the prevalence of early sexual intercourse among post-partum women and parity was evaluated by using four studies [21, 32, 33, 37]. The result revealed that the primipara was significantly associated with the early resumption of sexual intercourse among post-partum women. Women with primipara were 2.88 times more likely to engage in early sexual intercourse during the postpartum period than those multipara women (AOR = 2.88, 95%CI = 1.41, 5.89). No evidence of heterogeneity was found in any of the studies. (I-squared = 100.0%, p = 0.00.Hence, a random effect model was used (Fig 5).

The association between the prevalence of early resumption of sexual intercourse among post-partum women and having a history of severe genital injury on the last delivery was evaluated by using three studies [26, 32, 33]. The result revealed that the pooled effect of having no history of severe genital injury on the last delivery was significantly associated with the early

**Table 4. Meta-regression analysis based on year of publication and country.**

| Source of heterogeneity | Coefficient | Standard error | P value |
|---|---|---|---|
| Publication year | .975 | .0682 | .72 |
| Country | 1.019 | 0.120 | 0.16 |

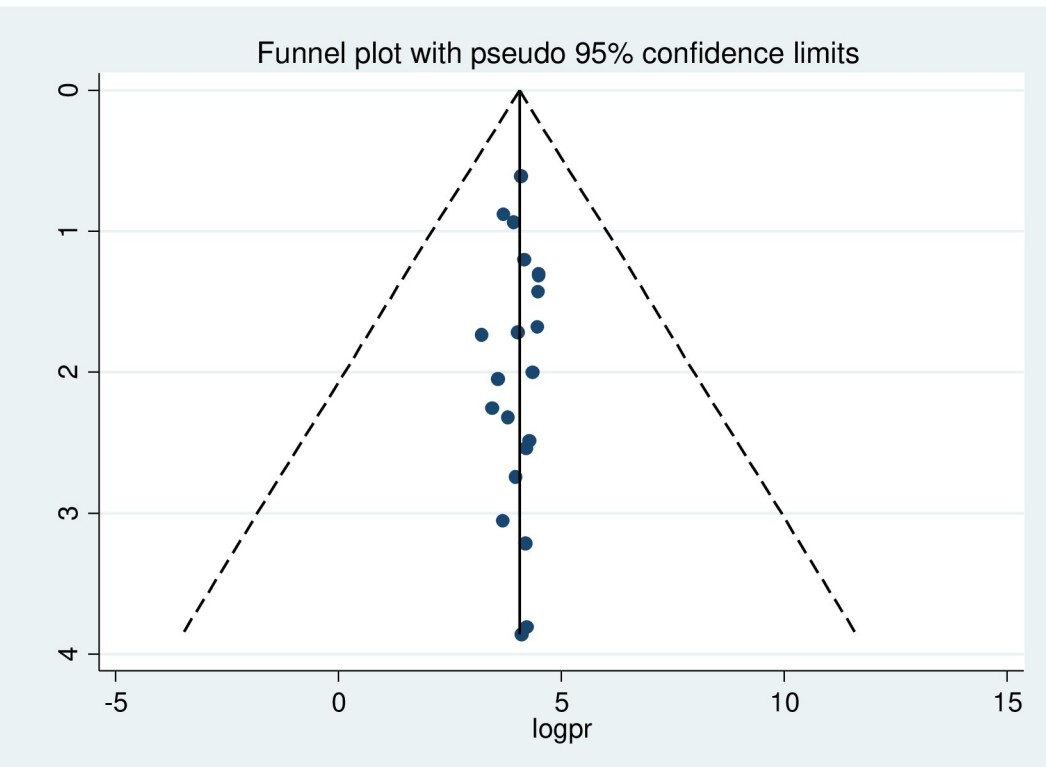

**Fig 3. Funnel plots for prevalence and factors associated with early resumption of sexual intercourse among postpartum women.**

resumption of sexual intercourse among post-partum women. Women who had no history of severe genital injury on the last delivery were 2.27 times more likely to have an early resumption of sexual intercourse than women who had a history of severe genital injury on the last delivery(AOR = 2.27, 95%CI = 1.05, 4.93). No evidence of heterogeneity was found in any of the studies (I-squared = 0.00%, p = 0.00) (Fig 6).

## Discussion

The World Health Organization has advised that research be done on sexual health because of its significance, separate from reproductive health, and because lack of knowledge about sexual health is the root cause of many dysfunctions and diseases throughout the world [42]. The compressive research findings must be included by healthcare professionals to implement evidence-based practice. Addressing the early resumption of postpartum intercourse and its factors is essential when focusing on postpartum issues of maternal health outcomes [43].

According to a database search, no systematic review has been conducted on the prevalence and factors associated with early resumption of sexual intercourse in postpartum women at a global level. Systematic reviews and meta-analyses are considered to provide the most robust evidence for clinical decision-making related to postpartum health compared to individual studies. This is due to the comprehensive and rigorous nature of these methods, allowing the synthesis and analysis of data from multiple studies. As a result, decision-makers are better equipped to achieve optimal postpartum sexual health outcomes.

In this systematic review and meta-analysis, the overall resumption of early sexual intercourse among post-partum was 57.26% (95% CI 50.14, 64.39). This review is lower than the

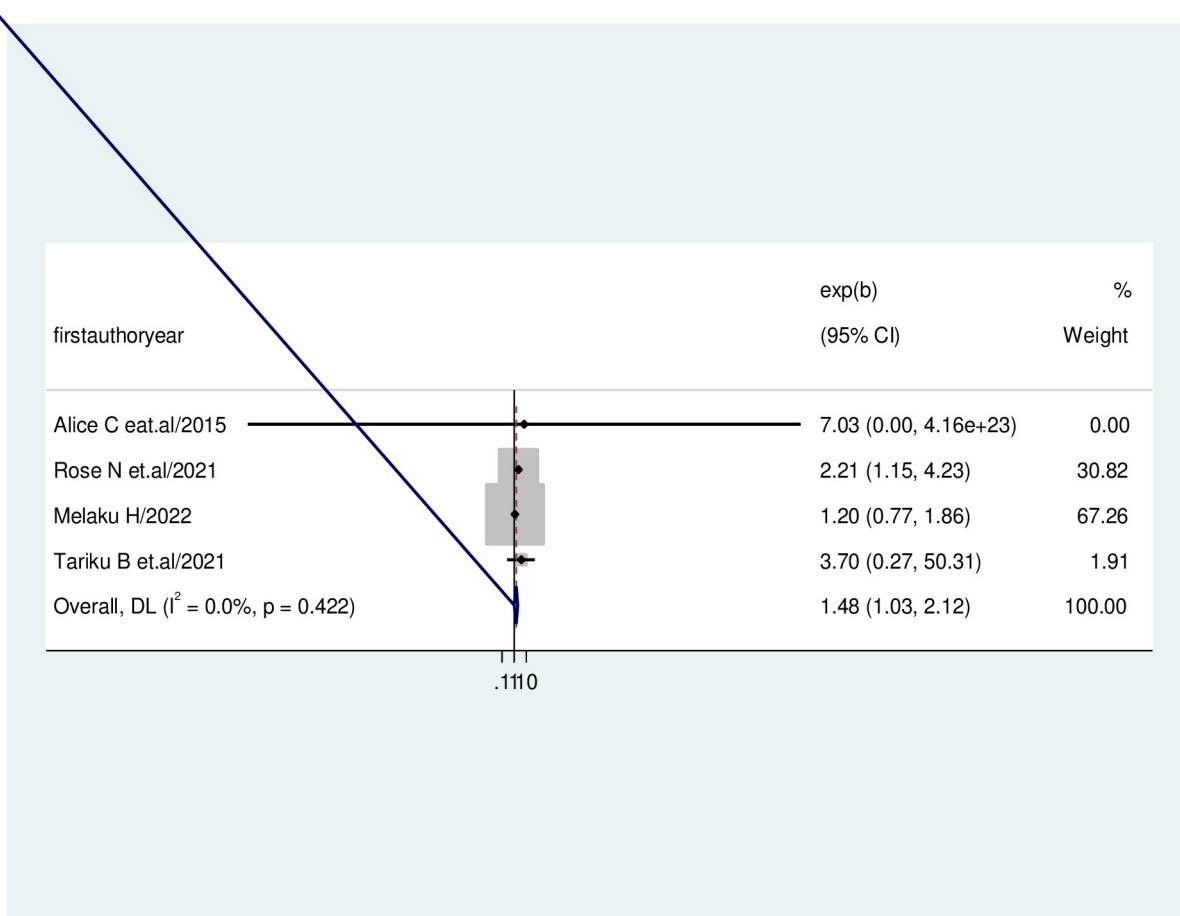

**Fig 4. Forest plot showing the association between the early resumption of sexual intercourse and current contraceptive usage.**

compared study in Sub-Saharan Africa [44]. These discrepancies may be explained by changes in sociocultural, social beliefs and norms, religious conduct, and the sexual attitudes of women in various geographic locations. Subjective norms, societal values, and ideas regarding postpartum sexual abstinence all have an impact on early sexual resumption [45].

For women from various cultural backgrounds, the postpartum period is known as a vulnerable and stressful time. During this time, women experienced enormous social and personal changes in addition to several new concerns and problems [46]. Given the significance of the first sexual encounter in establishing a committed relationship, postpartum sexual function is a significant concern for couples. Several factors, including lower parity, breastfeeding, cesarean section, severe genital injury, living child, maternal and paternal educational status, occupation, and current contraceptive use, could influence the early resumption of sexual intercourse in the postpartum period [47]. We found that the use of contraceptives; having less parity and having severe genital tears on the last delivery were statically associated with early resumption of sexual intercourse during the post-partum period. In the previous study, one of the factors linked to the early postpartum resumption of sexual activity was the use of contraceptives during the postpartum period [48]. Women who used contraceptives were 1.48 times more likely to have an early resumption of sexual intercourse during the postpartum period than those who didn't use (AOR = 1.48, 95%CI = 1.03, 6.2.12). This may be due to women who use contraceptives believing that they are not a risk for pregnancy, which encourages them to resume sexual activity six weeks after giving birth. However, early resumption of

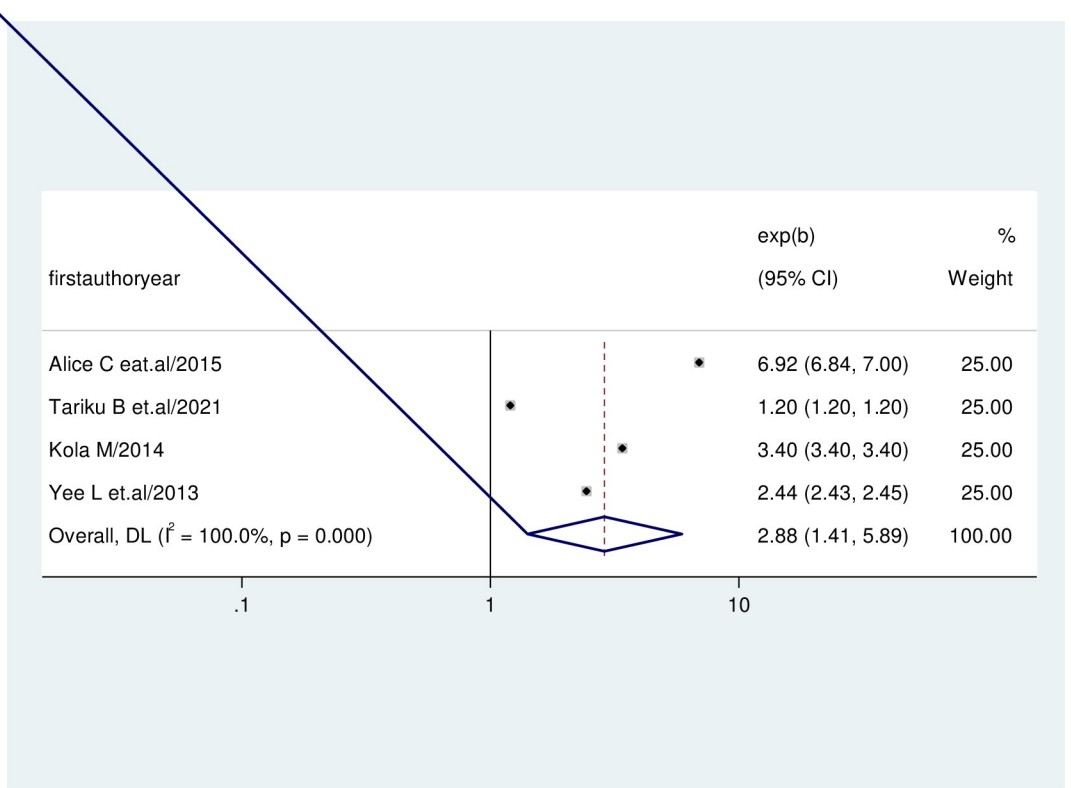

**Fig 5. Forest plot showing the association between the early resumption of sexual intercourse and parity.**

sexual activity may increase the risk of postpartum sexual dysfunction and unintended pregnancy [49]. Similarly, parity has affected women's early resumption of sexual intercourse; primiparous women are more likely to have an early resumption of sexual intercourse among postpartum women than multiparous women [50]. We found that women with primipara were 2.88 times more likely to engage in early sexual intercourse during the postpartum period than those multipara women (AOR = 2.88, 95%CI = 1.41, 5.89). This may be explained by their lack of experience, primiparae typically feel less confident in their postpartum sexual intercourse.

After giving birth, women who have trauma to the perineum may experience discomfort and other issues. First, second, third, and fourth-degree tears are used to characterize the degree of injury, with first-degree tears causing the least harm and fourth-degree tears causing the most. The anal sphincter or mucosa is affected by third- and fourth-degree tears, which are the most problematic [51]. Additionally, several perineal methods are employed to delay the head birth of the infant. Midwives and other delivery attendants frequently utilize massage, warm compresses, and various perineal management techniques. If these reduce trauma and discomfort for women, it is important to know. Even though episiotomy is one of the most frequently done procedures, there is an ongoing dispute in the professional literature about whether it has a preventive effect from third- and fourth-degree tears [52]. According to this review, no history of severe genital injury (severe perineal injury) on the last delivery had effects on the early resumption of sexual intercourse. Women who had no history of severe genital injury on the last delivery were 2.27 times more likely to have an early resumption of sexual intercourse than women who had a history of severe genital injury on the last delivery (AOR = 2.27, 95%CI = 1.05, 4.93). This might be the result of women who haven't experienced postpartum dyspareunia (painful sexual intercourse).

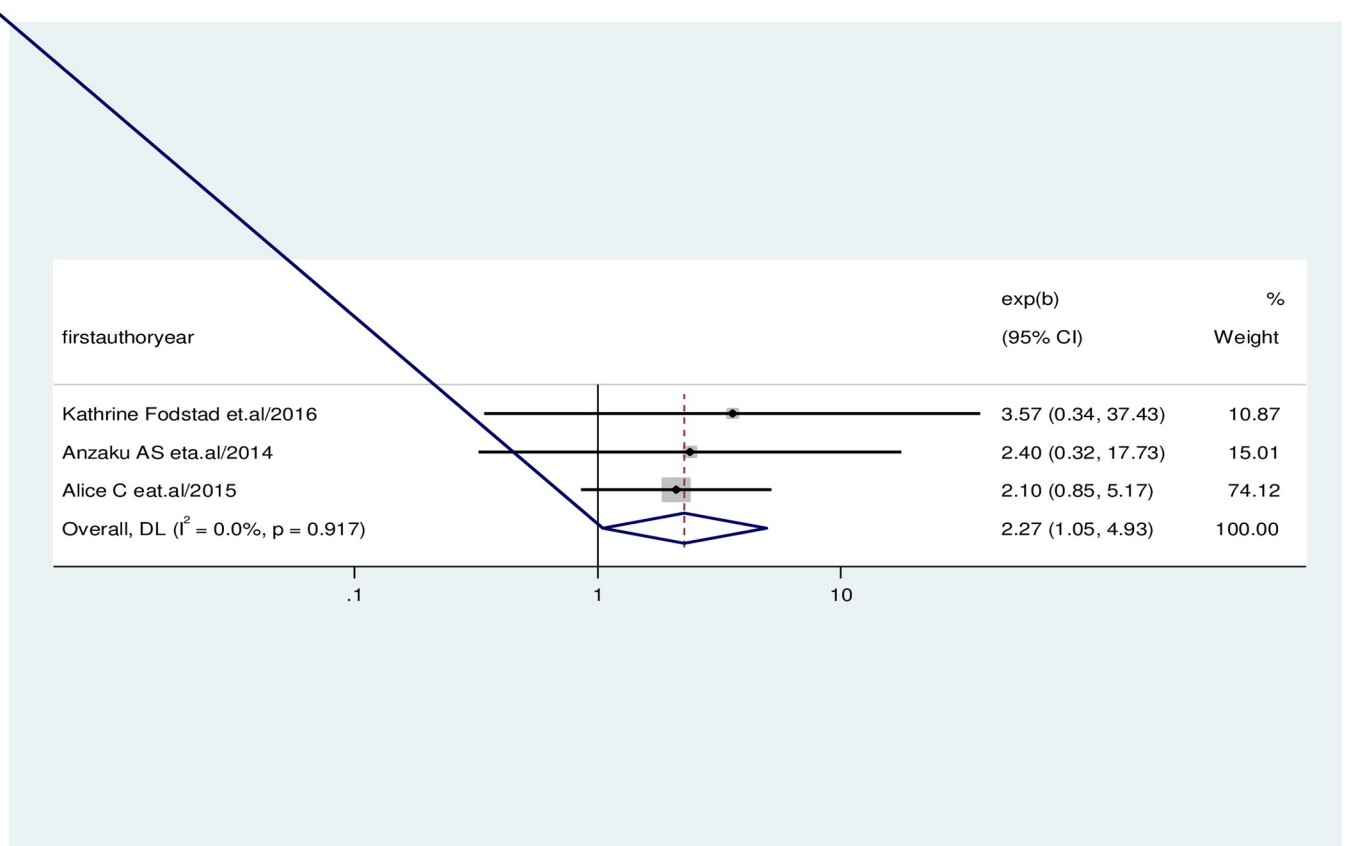

**Fig 6. Forest plot showing the association between the early resumption of sexual intercourse and a history of severe genital injury.**

### Strengths and limitations of the study

To reduce selection bias, we conducted a systematic literature review and included research based on clearly defined criteria. However, we only examined English-language publications. In addition, we included preprint articles that have not yet been peer-reviewed. The results of these studies may change in the future and there may be methodological biases.

### Conclusion

This systematic review and meta-analyses found that more than half of women resumed sexual intercourse early after giving birth. This suggests that a significant number of women may be at higher risk of unwanted pregnancies, short birth intervals, postpartum sepsis, and other factors. Contraceptive use, parity, and history of serious genital injury were significantly associated with early resumption of sexual intercourse in postpartum women. Therefore, stakeholders should improve the integration of postpartum sex education into maternal health services, and obstetricians should focus on these criteria while educating prenatal women about resuming sexual intercourse after childbirth.

### Supporting information

**S1 Table. This is the PRISMA 2020 checklist.**
(DOCX)

**S2 Table. This quality assessment for the 21 included studies.**
(DOCX)

## Author Contributions

**Conceptualization:** Kelemu Abebe Gelaw.

**Formal analysis:** Kelemu Abebe Gelaw, Yibeletal Assefa Atalay, Natnael Atnafu Gebeyehu.

**Investigation:** Kelemu Abebe Gelaw, Belete Gelaw Walle.

**Methodology:** Adisu Yeshambel, Getachew Asmare Adella.

**Resources:** Adisu Yeshambel, Liknaw Bewket Zeleke.

**Software:** Kelemu Abebe Gelaw, Yibeletal Assefa Atalay, Getachew Asmare Adella.

**Supervision:** Getachew Asmare Adella, Belete Gelaw Walle, Liknaw Bewket Zeleke.

**Validation:** Yibeletal Assefa Atalay, Adisu Yeshambel.

**Visualization:** Getachew Asmare Adella, Natnael Atnafu Gebeyehu.

**Writing – original draft:** Kelemu Abebe Gelaw, Yibeletal Assefa Atalay.

**Writing – review & editing:** Kelemu Abebe Gelaw, Liknaw Bewket Zeleke.

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
