## [Decision Letter · Decision Letter 0]

2 Jun 2023

PONE-D-23-11384Factors associated with early resumption of sexual intercourse among postpartum women: Systematic review and meta-analysisPLOS ONE

Dear Dr. Abebe,

Thank you for submitting your manuscript to PLOS ONE. After careful consideration, we feel that it has merit but does not fully meet PLOS ONE’s publication criteria as it currently stands. Therefore, we invite you to submit a revised version of the manuscript that addresses the points raised during the review process.

We look forward to receiving your revised manuscript.

Kind regards,

Antonio Simone Laganà, M.D., Ph.D.

Academic Editor

PLOS ONE

www.ajol.info

https://www.nature.com/articles/s41598-020-74477-z

In your revision ensure you cite all your sources (including your own works), and quote or rephrase any duplicated text outside the methods section. Further consideration is dependent on these concerns being addressed.

Additional Editor Comments:

The topic of the manuscript is interesting. Nevertheless, the reviewers raised several concerns: considering this point, I invite authors to perform the required major revisions.

Reviewers' comments:

Reviewer's Responses to Questions

**Comments to the Author**

1. Is the manuscript technically sound, and do the data support the conclusions?

Reviewer #1: Yes

Reviewer #2: Partly

Reviewer #3: Partly

Reviewer #4: No

2. Has the statistical analysis been performed appropriately and rigorously? 

Reviewer #1: Yes

Reviewer #2: Yes

Reviewer #3: No

Reviewer #4: No

3. Have the authors made all data underlying the findings in their manuscript fully available?

Reviewer #1: Yes

Reviewer #2: No

Reviewer #3: No

Reviewer #4: Yes

4. Is the manuscript presented in an intelligible fashion and written in standard English?

Reviewer #1: Yes

Reviewer #2: Yes

Reviewer #3: No

Reviewer #4: No

5. Review Comments to the Author

Reviewer #1: I read with great interest the Manuscript titled " Factors associated with early resumption of sexual intercourse among postpartum women: Systematic review and meta-analysis”, topic interesting enough to attract readers' attention.

Authors should clarify some point and improve the quality of manuscript citing relevant and novel key articles about the topic:

- I suggest a round of language revision, in order to correct few typos and improve readability.

- Authors should add further details to discuss the role of the perineum protection techniques during the management of the second phase of labour and the effect on the postpartum period (authors may refer to: PMID: 25909491; PMID: 24942141).

Because of these reasons, the article should be revised and completed. Tables and images are clear and interesting. Considering all these points, I think it could be of interest to the readers and, in my opinion, it deserves the priority to be published after minor revisions.

Reviewer #2: I suggest to add further details abot use and choice of contraception and to highlight the effect on different way of delivery (cesarean section vs vaginal delivery) on female post partum sexual functioning (author may refer to: PMID: 27318024, PMID: 24942141

Reviewer #3: Thank you for inviting me to review an interested topic of women’s health entitled “Factors associated with early resumption of sexual intercourse among postpartum women: Systematic review and meta-analysis”. The authors of this review tried to determine the global prevalence of early sexual resumption following birthing and estimate pooled effect size for common identified factors influencing it. Accordingly, I reviewed the manuscript and raised the following major concerns. Though there are many minor errors in the document that should be revised, I give emphasis and write here the major issues.

1. The authors didn’t provide justification or rationale why they conducted this review. Moreover, the objective of the study is not clearly stated as it was better to stated clear objective at the introduction part.

2. The inclusion and exclusion criteria are not clear. For instance, what the authors would like mean by “Studies were exclude (I read as included, but you better to change it) if they reported an observational study on the variables influencing the early resumption of sexual activity among postpartum women, described the techniques used to evaluate such activity,…..”. and “Studies were excluded if unrelated research works;…..”

3. The search strategy for each database with the number of articles identified should be provided as supplementary file.

4. What do you mean by ‘the search period was from February 15/2023 to February 29/2023”? It would be better to state here the last date of search that all databases were checked.

5. A total of 80,510 results identified as per the given searching term from the manuscript for PubMed: (((((("early resumption"[Text Word] OR "return"[Text Word]) AND "sexual intercourse"[Text Word]) OR "Coitus"[MeSH Terms]) AND ("Factor"[Text Word] OR "determinants"[Text Word])) OR "Risk factors"[MeSH Terms]) AND ("Postpartum"[Text Word] OR "post-delivery"[Text Word])) OR "Postpartum period"[MeSH Terms]. I checked it by considering the last date of search was February 28/2023 because February never be 29 (though your search period indicates up to February 29/2023). However, you stated that 7,228 records identified through PubMed database searching (Figure 1). How this much variation is there?

6. As per described in supplementary file 3, the quality all studies was assessed with similar components which is not acceptable because the JBI tool had different components for each study design. Also, what is the need of reporting supplementary file 2 and 3 which both are quality assessment results (supplementary file 2 sounds good than 3)

7. In Figure 1, the number of articles in each database and other additional sources should be recorded per each source. However, you didn’t to record the search terms used for and number of articles identified in each database. This reduces the trustworthiness of your searching strategy and number of articles identified.

8. The authors stated that “an extensive data search was performed on PubMed, Web of Science, Scopus, Google Scholar, Cochrane Library, and African Journals Online (AJOL) databases used to get the research articles”. However, where are the results of Web of Science and Cochrane Library? And how was Scopus, Web of Science and Cochrane Library? Furthermore, Scopus is a database that shouldn’t be included with other sources (Figure 1).

9. For study screening and selection process (Figure 1), you should use the ‘PRISMA 2020 flow diagram for new systematic reviews which included searches of databases, registers and other sources’, which is available at: http://www.prisma-statement.org/PRISMAStatement/FlowDiagram.

10. All 79 articles which were excluded after the examination of their full text should be either cited in the manuscript or should be included as supplementary file with reason. The current systematic review and the PRISMA checklist that you have used strongly recommended it. In the PRISMA checklist that you have used, you stated that “not applicable”; why for? it is belied that you screened all the 79 articles and excluded with reason. In addition, The appropriate PRISMA checklist without no need of edition is available in both PDF and Word doc at: http://www.prisma-statement.org/PRISMAStatement/Checklist.aspx.

11. The other major issue I observed is pooled proportion and cases together for meta-analysis. How is possible to pooled cases in case control study with proportions of cohort and cross-sectional studies. Where did you get the proportion/prevalence for case control studies because cases are not proportion. This totally produces a misleading and unacceptable result.

12. The reported prevalence among studies ranged from 20.2 to 90.2 %, which is very wide variation, and which is difficult to pooled together. However, you did a meta-analysis using random effects model even without acknowledging it.

13. I strongly recommended you revising your literature searching and include the many studies which were missing to be included in your review; then execute the analysis. Besides, better to review the document again and again before submitting to the journal.

14. Extensive grammar errors are observed throughout the manuscript with many statements are difficult to understand. Thus, thoroughly rereading and rewriting or language consultation might be necessary.

15. Though the author stated that they used the reference manager endnote software, the reference lists don’t seem Endnote output. Better you check it.

16. The final critical issue is that the manuscript has very high (46%) textual similarity with existing studies. This ithenticate report couldn’t include references.

Reviewer #4: Review comments by Mohammed S. Obsa

Factors associated with early resumption of sexual intercourse among postpartum.

women: Systematic review and meta-analysis

Write a step-by-step response to these comments:

•

• The area of the study is very important; however, the manuscript needs major revision to be accepted for publications in the PLoS one.

• It is essential that a manuscript should undergo gross language editing before it is accepted for publication in PLoS one.

• Include the total sample size of this study in the abstract as well.

• Explain briefly why you used statistical methods in your abstracts for major findings.

• This conclusion does not seem to make sense, so it should be refined.

• Would you recommend resuming early sexual relations?

• The introduction should begin with a brief description of the study's background. The introduction in this case was not focused.

• In the introduction, the author should describe the magnitude of the problem and what factors affect it.

• This gap was not clearly identified by the author, and it would be helpful if they could specify where it lies?

• There should be a thorough mention of the key term used for the search. The manuscript presents search terms inconsistently.

• In both cases, you mention exclusion criteria, but they are contradictory. It is therefore necessary to revise it thoroughly. It is recommended that the exclusion criteria be the default inclusion criteria.

• The author should use a citation software programme like Endnote. It is evident from what is written in the characteristics of the included studies that poor citation styles have been applied.

• The order in which the results were presented was inappropriate. After exploring sources of heterogeneity, publication bias should be investigated.

· The author should check the assumptions for subgroup analysis before running the data.

· The discussion should be substantially revised. There is a lack of coherence and implications of the major findings in the paper.

• It is recommended that the order of discussion follow the order of importance of the variables.

• There should be a clear explanation of the strengths and limitations of this review.

6. PLOS authors have the option to publish the peer review history of their article (what does this mean?). If published, this will include your full peer review and any attached files.

Reviewer #1: **Yes: **Ilaria Cuccu

Reviewer #2: No

Reviewer #3: **Yes: **Birye Dessalegn Mekonnen

Reviewer #4: **Yes: **Mohammed Suleiman Obsa

---

## [Author Response · Author response to Decision Letter 0]

15 Jun 2023

Reviewer 1

Reviewer #1: I read with great interest the Manuscript titled " Factors associated with early resumption of sexual intercourse among postpartum women: Systematic review and meta-analysis”, topic interesting enough to attract readers' attention.

Authors should clarify some point and improve the quality of manuscript citing relevant and novel key articles about the topic:

- I suggest a round of language revision, in order to correct few typos and improve readability.

- Authors should add further details to discuss the role of the perineum protection techniques during the management of the second phase of labour and the effect on the postpartum period (authors may refer to: PMID: 25909491; PMID: 24942141).

Author response: After giving birth, women who have trauma to the perineum may experience discomfort and other issues. First, second, third, and fourth degree tears are used to characterize the degree of injury, with first degree tears causing the least harm and fourth degree tears causing the most. The anal sphincter or mucosa is affected by third- and fourth-degree tears, which are the most problematic(49). Additionally, several perineal methods are employed to delay the head birth of the infant. Midwives and other delivery attendants frequently utilize massage, warm compresses, and various perineal management techniques. If these reduce trauma and discomfort for women, it is important to know. Even though episiotomy is one of the most frequently done procedures, there is an ongoing dispute in the professional literature about whether it has a preventive effect from third- and fourth-degree tears(50).

Because of these reasons, the article should be revised and completed. Tables and images are clear and interesting. Considering all these points, I think it could be of interest to the readers and, in my opinion, it deserves the priority to be published after minor revisions.

Author response: We accepted your comments and suggestions.

Reviewer 2

Reviewer #2: I suggest to add further details abot use and choice of contraception and to highlight the effect on different way of delivery (cesarean section vs vaginal delivery) on female post partum sexual functioning (author may refer to: PMID: 27318024, PMID: 24942141

Author response:we didn’t acces it.But we were discuss with other similar literature.

Reviewer 3

Reviewer #3: Thank you for inviting me to review an interested topic of women’s health entitled “Factors associated with early resumption of sexual intercourse among postpartum women: Systematic review and meta-analysis”. The authors of this review tried to determine the global prevalence of early sexual resumption following birthing and estimate pooled effect size for common identified factors influencing it. Accordingly, I reviewed the manuscript and raised the following major concerns. Though there are many minor errors in the document that should be revised, I give emphasis and write here the major issues.

1. The authors didn’t provide justification or rationale why they conducted this review. Moreover, the objective of the study is not clearly stated as it was better to stated clear objective at the introduction part.

Author response: we have corrected as follows

Hence, this study aimed to assess the prevalence and factors associated with early resuming postpartum sexual intercourse at the global level.

2. The inclusion and exclusion criteria are not clear. For instance, what the authors would like mean by “Studies were exclude (I read as included, but you better to change it) if they reported an observational study on the variables influencing the early resumption of sexual activity among postpartum women, described the techniques used to evaluate such activity,…..”. and “Studies were excluded if unrelated research works;…..”

Author response: we have corrected as follows

Only studies that had full-text articles, English language articles, published and unpublished articles, cross-sectional, case-control and cohort research were included in this study. However, duplicate sources, interventional studies, case reports, systematic reviews, qualitative articles, case series, conference abstracts, letters to editors, and any articles that were not fully accessible after at least two emails had been exchanged with the primary author were all excluded. The COCOPOP(Condition, Context, and Population) paradigm was utilized to determine the suitability of the included studies for this investigation. Postpartum women who had early resumption of sexual intercourse made up the study's population (POP), while the prevalence of early resumption of sexual intercourse served as the condition and global served as the setting.

3. The search strategy for each database with the number of articles identified should be provided as supplementary file.

Author response:accepted

4. What do you mean by ‘the search period was from February 15/2023 to February 29/2023”? It would be better to state here the last date of search that all databases were checked.

Author response: we have corrected as follows

The last date of search that all databases were checked from February 15/2023 to February 29/2023.

5. A total of 80,510 results identified as per the given searching term from the manuscript for PubMed: (((((("early resumption"[Text Word] OR "return"[Text Word]) AND "sexual intercourse"[Text Word]) OR "Coitus"[MeSH Terms]) AND ("Factor"[Text Word] OR "determinants"[Text Word])) OR "Risk factors"[MeSH Terms]) AND ("Postpartum"[Text Word] OR "post-delivery"[Text Word])) OR "Postpartum period"[MeSH Terms]. I checked it by considering the last date of search was February 28/2023 because February never be 29 (though your search period indicates up to February 29/2023). However, you stated that 7,228 records identified through PubMed database searching (Figure 1). How this much variation is there?

Author response:accepted and checked 

6. As per described in supplementary file 3, the quality all studies was assessed with similar components which is not acceptable because the JBI tool had different components for each study design. Also, what is the need of reporting supplementary file 2 and 3 which both are quality assessment results (supplementary file 2 sounds good than 3)

Author response:accepted and changed

7. In Figure 1, the number of articles in each database and other additional sources should be recorded per each source. However, you didn’t to record the search terms used for and number of articles identified in each database. This reduces the trustworthiness of your searching strategy and number of articles identified.

Author response: We have mentioned in The PRISMA diagram

8. The authors stated that “an extensive data search was performed on PubMed, Web of Science, Scopus, Google Scholar, Cochrane Library, and African Journals Online (AJOL) databases used to get the research articles”. However, where are the results of Web of Science and Cochrane Library? And how was Scopus, Web of Science and Cochrane Library? Furthermore, Scopus is a database that shouldn’t be included with other sources (Figure 1).

Author response:We have mentioned in The PRISMA diagram

9. For study screening and selection process (Figure 1), you should use the ‘PRISMA 2020 flow diagram for new systematic reviews which included searches of databases, registers and other sources’, which is available at: http://www.prisma-statement.org/PRISMAStatement/FlowDiagram.

Author response: corrected as follow

Fig 1: PRISMA flow chart displays studies used for Systematic Review and Meta-analysis of Prevalence and factors associated with early resumption of sexual intercourse among postpartum women 

10. All 79 articles which were excluded after the examination of their full text should be either cited in the manuscript or should be included as supplementary file with reason. The current systematic review and the PRISMA checklist that you have used strongly recommended it. In the PRISMA checklist that you have used, you stated that “not applicable”; why for? it is belied that you screened all the 79 articles and excluded with reason. In addition, The appropriate PRISMA checklist without no need of edition is available in both PDF and Word doc at: http://www.prisma-statement.org/PRISMAStatement/Checklist.aspx.

Author response: We updated with current PRISMA checklist

11. The other major issue I observed is pooled proportion and cases together for meta-analysis. How is possible to pooled cases in case control study with proportions of cohort and cross-sectional studies. Where did you get the proportion/prevalence for case control studies because cases are not proportion. This totally produces a misleading and unacceptable result.

Author response: Accepted 

12. The reported prevalence among studies ranged from 20.2 to 90.2 %, which is very wide variation, and which is difficult to pooled together. However, you did a meta-analysis using random effects model even without acknowledging it.

Author response: Accepted 

13. I strongly recommended you revising your literature searching and include the many studies which were missing to be included in your review; then execute the analysis. Besides, better to review the document again and again before submitting to the journal.

Author response: we revised again the whole document

14. Extensive grammar errors are observed throughout the manuscript with many statements are difficult to understand. Thus, thoroughly rereading and rewriting or language consultation might be necessary.

Author response: we have revised and corrected any grammar errors, spelling, active/passive voice, and others. Additionally, we edited through Grammarly

15. Though the author stated that they used the reference manager endnote software, the reference lists don’t seem Endnote output. Better you check it.

Author response: admmited

16. The final critical issue is that the manuscript has very high (46%) textual similarity with existing studies. This ithenticate report couldn’t include references.

Author response: We have paraphrased it again.

Reviewer 4

 Reviewer #4: Review comments by Mohammed S. Obsa

Factors associated with early resumption of sexual intercourse among postpartum.

women: Systematic review and meta-analysis

Write a step-by-step response to these comments:

• The area of the study is very important; however, the manuscript needs major revision to be accepted for publications in the PLoS one.

• It is essential that a manuscript should undergo gross language editing before it is accepted for publication in PLoS one.

1. Include the total sample size of this study in the abstract as well.

 Author response: twenty-one studies with 4,482 postpartum women participants were included in the Systematic Review and Meta-analysis

2. Explain briefly why you used statistical methods in your abstracts for major findings.

Author response: We have revised as follows

Methods: Searches were conducted in PubMed, Web of Science, Science Direct, Google Scholar, African Journals Online, and the Cochrane Library. Data were extracted using Microsoft Excel, and STATA version 14 was used for analysis. Publication bias was checked by funnel plot, Egger, and Begg regression tests. A p-value of 0.05 was regarded to indicate potential publication bias. Using I2 statistics, the heterogeneity of the studies was evaluated. By country, a subgroup analysis was conducted. A sensitivity analysis was carried out to determine the effect of each study's findings on the overall estimate. The random effect model was used to assess the overall effect of the study, then measured by the prevalence rates and odds ratio with 95% CI. The result was presented in the form of text, tables, and figures.

3. This conclusion does not seem to make sense, so it should be refined.

Author response: We have corrected as follows

 This study revealed that more than half of postpartum women had early resumption of sexual intercourse. This suggests that a significant number of women may be at higher risk for high rates of unwanted pregnancy, short birth interval, and postpartum sepsis. Thus, stakeholders should improve the integration of postpartum sexual education with maternal health services to reduce the resumption of postpartum sexual intercourse.

4. Would you recommend resuming early sexual relations?

Author response: No, Early resumption of sexual intercourse among postpartum women may cause maternal health problems: unfavorable birth outcomes, wound infection, painful intercourse, vaginal dryness, and inability to achieve orgasm (11). In addition, early resuming sexual intercourse without utilizing contraception after giving birth can lead to an unplanned pregnancy and short birth interval. Due to this, there may be a high risk of death for infants under the age of one year (12).

5. The introduction should begin with a brief description of the study's background. The introduction in this case was not focused.

Author response: We have revised as follows

Sexuality evolves throughout a person's life and is essential to being human (1). It is a key idea in human sexual health (2). It has at least three connected purposes: communication, pleasure, and reproduction. Pregnancy, birth, and postpartum are all critical times for sexual health (4). Women's bodies experience changes after giving birth that have an impact on both their health and other parts of their lives, including sexual activity, which is essential to human existence(5). Postpartum sexual health is indicated by resuming sexual activity as well as by arousal, desire, orgasm, and sexual satisfaction (6). World Health Organization (WHO) recognizes postpartum sexual health to be one of the most crucial issues that need to be addressed during the postpartum period(7). 

 The desire to engage in sexual activity declines during pregnancy but returns to normal, usually six weeks after delivery (8). Resuming postpartum sexual activity should be done at least six weeks after giving birth(9). However, the decision to resume sexual activity after delivery differs from woman to woman and is influenced by several factors such as the quantity of bleeding, mode of delivery, culture, maternal mental health, infant health, the relationship with one's partner, and the mother's general health (10).

 Early resumption of sexual intercourse among postpartum women may cause maternal health problems: unfavorable birth outcomes, wound infection, painful intercourse, vaginal dryness, and inability to achieve orgasm (11). In addition, early resuming sexual intercourse without utilizing contraception after giving birth can lead to an unplanned pregnancy and short birth interval. Due to this, there may be a high risk of death for infants under the age of one year (12). Sociocultural norms and beliefs, education, the place of the birth, the mother's breastfeeding status, mode of delivery, low parity, using contraceptives, and residence are some of the factors that have been found as having an effect on the early resumption of sexual activity after delivery (13). 

 Even though the WHO advises that all women should be evaluated 2–6 weeks after childbirth, the problem of early resumption of sexual activity during postpartum has received little attention from researchers, policymakers, and healthcare providers (14). Previous studies reported that exclusively address the utilization of postpartum family planning in the early postpartum period rather than identifying the risk factors for an early resumption of sexual intercourse among post-partum women (15, 16). As a result, it's vital to study the various aspects of the early resumption of sexual relationships during the postpartum period to understand the direct and indirect effects of this problem and emphasize the significance of looking into these issues(17). Furthermore, the prevalence of early resumption of sexual intercourse is varying in the previous studies, with rates ranging from 20.2% in Ethiopia (39) to 90.2% in the USA (21). Given these variances, there is no general estimation of early resumption of sexual intercourse among postpartum women at the global level. The present study aimed to ascertain the overall prevalence of early resumption of sexual intercourse among most women in the world.

 Prevalence and factors contributing to the early resumption of postpartum sexual intercourse at the global level also require further study. Additionally, there was no systematic review or meta-analysis conducted on the prevalence and contributing variables associated with early postpartum sexual resumption worldwide. Identifying factors that impact on early resumption of sexual activity to lessen or eliminate issues and considerably improve the well-being of postpartum women. Additionally, the results of this study might help in developing new approaches to postpartum sexuality education. Hence, this study aimed to assess the prevalence and factors associated with early resuming postpartum sexual intercourse at the global level.

6. In the introduction, the author should describe the magnitude of the problem and what factors affect it.

Author response: the prevalence of early resumption of sexual intercourse is varying in the previous studies, with rates ranging from 20.2% in Ethiopia (39) to 90.2% in the USA (21). Sociocultural norms and beliefs, education, the place of the birth, the mother's breastfeeding status, mode of delivery, low parity, using contraceptives, and residence are some of the factors that have been found as having an effect on the early resumption of sexual activity after delivery (13). 

7. This gap was not clearly identified by the author, and it would be helpful if they could specify where it lies?

Author response: Previous studies reported that exclusively address the utilization of postpartum family planning in the early postpartum period rather than identifying the risk factors for an early resumption of sexual intercourse among post-partum women (15, 16). As a result, it's vital to study the various aspects of the early resumption of sexual relationships during the postpartum period to understand the direct and indirect effects of this problem and emphasize the significance of looking into these issues(17). Furthermore, the prevalence of early resumption of sexual intercourse is varying in the previous studies, with rates ranging from 20.2% in Ethiopia (39) to 90.2% in the USA (21). Given these variances, there is no general estimation of early resumption of sexual intercourse among postpartum women at the global level. The present study aimed to ascertain the overall prevalence of early resumption of sexual intercourse among most women in the world.

 Prevalence and factors contributing to the early resumption of postpartum sexual intercourse at the global level also require further study. Additionally, there was no systematic review or meta-analysis conducted on the prevalence and contributing variables associated with early postpartum sexual resumption worldwide. Identifying factors that impact on early resumption of sexual activity to lessen or eliminate issues and considerably improve the well-being of postpartum women. Additionally, the results of this study might help in developing new approaches to postpartum sexuality education. Hence, this study aimed to assess the prevalence and factors associated with early resuming postpartum sexual intercourse at the global level.

• There should be a thorough mention of the key term used for the search. The manuscript presents search terms inconsistently.

8. In both cases, you mention exclusion criteria, but they are contradictory. It is therefore necessary to revise it thoroughly. It is recommended that the exclusion criteria be the default inclusion criteria.

Author response: we have corrected as follows

Eligibility criteria

 Studies that had full-text articles, English language articles, both published and unpublished articles, cross-sectional studies, case-control studies, and cohort studies were included in this study. Duplicate sources, interventional studies, case reports, systematic reviews, qualitative articles, case series, conference abstracts, letters to editors, and any articles that were not fully accessible after at least two emails had been exchanged with the primary author were all excluded. A COCOPOP(Condition, Context, and Population) paradigm was utilized to determine the suitability of the included studies for this investigation. Postpartum women who had early resumption of sexual intercourse made up the study's population (POP), while the prevalence of early resumption of sexual intercourse served as the condition and global served as the setting.

9.The author should use a citation software programme like Endnote. It is evident from what is written in the characteristics of the included studies that poor citation styles have been applied.

Author response: accepted.

10. The order in which the results were presented was inappropriate. After exploring sources of heterogeneity, publication bias should be investigated.

Author response: accepted.

• The author should check the assumptions for subgroup analysis before running the data.

• The discussion should be substantially revised. There is a lack of coherence and implications of the major findings in the paper.

11. Author response: We have revised as follows

Discussion

 The World Health Organization has advised that research be done on sexual health because of its significance, separate from reproductive health, and because lack of knowledge about sexual health is the root cause of many dysfunctions and diseases throughout the world(41). The compressive research findings must be included by healthcare professionals to implement evidence-based practice. Addressing the early resumption of postpartum intercourse and its factors is essential when focusing on postpartum issues of maternal health outcomes (42).

 According to a database search, no Systematic Review has been conducted on the prevalence and factors associated with early resumption of sexual intercourse among postpartum women at a global level. Systematic reviews and meta-analyses have the best evidence for clinical judgment than individual studies. In this systematic review and meta-analysis, the overall resumption of early sexual intercourse among post-partum was 57.26% (95% CI 50.14, 64.39). The current study is lower than the one conducted in Sub-Saharan Africa(43). These discrepancies may be explained by changes in sociocultural, social beliefs and norms, religious conduct, and the sexual attitudes of women in various geographic locations. Subjective norms, societal values, and ideas regarding postpartum sexual abstinence all have an impact on early sexual resumption(44).

 For women from various cultural backgrounds, the postpartum period is known as a vulnerable and stressful time. During this time, women experience enormous social and personal changes in addition to several new concerns and problems(45). Given the significance of the first sexual encounter in establishing a committed relationship, postpartum sexual function is a significant concern for couples. Several factors, including less parity, breastfeeding, cesarean section, severe genital injury, alive child, maternal and paternal educational status, occupation, and current uses of contraceptives might affect the early resumption of sexual intercourse during the postpartum period(46).

 We found that the use of contraceptives; having less parity and having severe genital tears on the last delivery were statically associated with early resumption of sexual intercourse during the post-partum period. According to a previous study, one of the factors linked to the early postpartum resumption of sexual activity is the use of contraceptives during the postpartum period(47). Women who used contraceptives were 1.48 times more likely to have an early resumption of sexual intercourse during the postpartum period than those who didn't use (AOR = 1.48, 95%CI = 1.03, 6.2.12). This may be the result of the fact that women who use contraceptives believe that they are not a risk for pregnancy, which encourages them to resume sexual activity six weeks after giving birth. However, early resumption of sexual activity may increase the risk of postpartum sexual dysfunction and unintended pregnancy (48).

 Similarly, parity has affected how women's early resumption of sexual intercourse; primiparous women are more likely to have an early resumption of sexual intercourse among postpartum women than multiparous women(49). We found that women with primipara were 2.88 times more likely to engage in early sexual intercourse during the postpartum period than those multipara women (AOR = 2.88, 95%CI = 1.41, 5.89). This may be explained by their lack of experience, primiparae typically feel less confident in their postpartum sexual intercourse.

 Even though episiotomy is one of the most frequently done procedures, there is an ongoing dispute in the professional literature about whether it has a preventive effect from third- and fourth-degree tears(50). According to this review, no history of severe genital injury(severe perineal injury) on the last delivery had effects on the early resumption of sexual intercourse. Women who had no history of severe genital injury on the last delivery were 2.27 times more likely to have an early resumption of sexual intercourse than women who had a history of severe genital injury on the last delivery(AOR = 2.27, 95%CI = 1.05, 4.93). This might be the result of women who haven't experienced postpartum dyspareunia (painful sexual intercourse). 

12. It is recommended that the order of discussion follow the order of importance of the variables.

Author response: admitted

13. There should be a clear explanation of the strengths and limitations of this review.

Author response: accepted and changed

---

## [Decision Letter · Decision Letter 1]

15 Sep 2023

PONE-D-23-11384R1Prevalence and factors associated with early resumption of sexual intercourse among postpartum women: Systematic Review and Meta-AnalysisPLOS ONE

Dear Dr. Abebe,

Thank you for submitting your manuscript to PLOS ONE. After careful consideration, we feel that it has merit but does not fully meet PLOS ONE’s publication criteria as it currently stands. Therefore, we invite you to submit a revised version of the manuscript that addresses the points raised during the review process. Please submit your revised manuscript by Oct 30 2023 11:59PM. If you will need more time than this to complete your revisions, please reply to this message or contact the journal office at plosone@plos.org. Please include the following items when submitting your revised manuscript:A rebuttal letter that responds to each point raised by the academic editor and reviewer(s). You should upload this letter as a separate file labeled 'Response to Reviewers'.A marked-up copy of your manuscript that highlights changes made to the original version. You should upload this as a separate file labeled 'Revised Manuscript with Track Changes'.An unmarked version of your revised paper without tracked changes. You should upload this as a separate file labeled 'Manuscript'.If applicable, we recommend that you deposit your laboratory protocols in protocols.io to enhance the reproducibility of your results. Protocols.io assigns your protocol its own identifier (DOI) so that it can be cited independently in the future. For instructions see: https://journals.plos.org/plosone/s/submission-guidelines#loc-laboratory-protocols. Additionally, PLOS ONE offers an option for publishing peer-reviewed Lab Protocol articles, which describe protocols hosted on protocols.io. Read more information on sharing protocols at https://plos.org/protocols?utm_medium=editorial-email&utm_source=authorletters&utm_campaign=protocols.

We look forward to receiving your revised manuscript.

Kind regards,

Frank T. Spradley

Academic Editor

PLOS ONE

Journal Requirements:

**Additional Editor Comments:**

Thank you for the resubmission of your study, which the reviewers still read with great interest. We thank the reviewers for their time and contributions to strengthen this body of work. Indeed, there is 1 issue that must be addressed pertaining to the current references cited. Please include updates references that better highlight the gap in knowledge that necessitated conducting this study and the importance of perineum protection techniques during the timing of management during labor.

Reviewers' comments:

Reviewer's Responses to Questions

**Comments to the Author**

1. If the authors have adequately addressed your comments raised in a previous round of review and you feel that this manuscript is now acceptable for publication, you may indicate that here to bypass the “Comments to the Author” section, enter your conflict of interest statement in the “Confidential to Editor” section, and submit your "Accept" recommendation.

Reviewer #1: All comments have been addressed

Reviewer #2: All comments have been addressed

2. Is the manuscript technically sound, and do the data support the conclusions?

Reviewer #1: Yes

Reviewer #2: Yes

3. Has the statistical analysis been performed appropriately and rigorously? 

Reviewer #1: Yes

Reviewer #2: I Don't Know

4. Have the authors made all data underlying the findings in their manuscript fully available?

Reviewer #1: Yes

Reviewer #2: Yes

5. Is the manuscript presented in an intelligible fashion and written in standard English?

Reviewer #1: Yes

Reviewer #2: Yes

6. Review Comments to the Author

Reviewer #1: 

I think that Authors should clarify some points and improve the quality of manuscript citing relevant and novel key articles about the topic:

-the role of the perineum protection techniques during the management of the second phase of labour and the effect on the postpartum period (authors may refer to: PMID: 25909491; PMID: 24942141).

I think that this modification make the work more complete. . Considered all these points, I think it could be of interest for the readers and, in my opinion, it deserves the priority to be published after minor revisions.

Reviewer #2: I carefully evaluated the revised version of this manuscript.

Authors have performed the required changes, improving significantly the quality of the paper.

7. PLOS authors have the option to publish the peer review history of their article (what does this mean?). If published, this will include your full peer review and any attached files.

Reviewer #1: No

Reviewer #2: **Yes: **Ornella Sgro

---

## [Author Response · Author response to Decision Letter 1]

17 Sep 2023

Point by point response for reviewer and edittor

Reviewer #1: 

I think that Authors should clarify some points and improve the quality of manuscript citing relevant and novel key articles about the topic:

-the role of the perineum protection techniques during the management of the second phase of labour and the effect on the postpartum period (authors may refer to: PMID: 25909491; PMID: 24942141).

I think that this modification make the work more complete. . Considered all these points, I think it could be of interest to the readers and, in my opinion, it deserves the priority to be published after minor revisions.

Author response: we have updated as following

 The previous study has provided evidence that the use of hands-free perineal control techniques during the second stage of labor may represent a promising delivery approach to maintain perineal integrity. In addition, these techniques can have a positive effect on the resumption of sexual activity in the postpartum period, as they include several factors that can facilitate this process [14]. Numerous cohort studies have shown that women who undergo spontaneous vaginal delivery with intact perineum have a higher likelihood of having vaginal intercourse again within six to eight weeks after birth compared to women who undergo an episiotomy assisted vaginal birth etc. undergo a Cesarean Section[15,16,17]. 

References 

14.Laganà AS, Burgio MA, Retto G, Pizzo A, Granese R, Sturlese E, Ciancimino L, Chiofalo B, Retto A, Triolo O. Management of the second phase of labour: perineum protection techniques. Minerva Ginecol. 2015 Jun 1;67(3):289-96.

15. Laganà AS, Burgio MA, Ciancimino L, Sicilia A, Pizzo A, Magno C, Butticè S, Triolo O. Evaluation of recovery and quality of sexual activity in women during postpartum in relation to the different mode of delivery: a retrospective analysis. Minerva Ginecol. 2015 Aug 1;67(4):315-20.

16. McDonald, E.; Brown, S. Does the method of birth make a difference to when women resume sex after childbirth? BJOG Int. J. Obstet. Gynaecol. 2013, 120, 823–830. [Google Scholar] [CrossRef]

Additional Editor Comments:

Thank you for the resubmission of your study, which the reviewers still read with great interest. We thank the reviewers for their time and contributions to strengthen this body of work. Indeed, there is 1 issue that must be addressed pertaining to the current references cited. Please include updates references that better highlight the gap in knowledge that necessitated conducting this study and the importance of perineum protection techniques during the timing of management during labor.

Author response: We have updated 

 The previous study has provided evidence that the use of hands-free perineal control techniques during the second stage of labor may represent a promising delivery approach to maintain perineal integrity. In addition, these techniques can have a positive effect on the resumption of sexual activity in the postpartum period, as they include several factors that can facilitate this process [14]. Numerous cohort studies have shown that women who undergo spontaneous vaginal delivery with intact perineum have a higher likelihood of having vaginal intercourse again within six to eight weeks after birth compared to women who undergo an episiotomy assisted vaginal birth etc. undergo a Cesarean Section[15,16,17]. 

Therefore, it is important to examine the various aspects of the early resumption of sexual relationships in the postpartum period to understand the direct and indirect effects of this problem and highlight the importance of studying these issues at a global level [19].

---

## [Editor Report · Decision Letter 2]

25 Oct 2023

Prevalence and factors associated with early resumption of sexual intercourse among postpartum women: Systematic Review and Meta-Analysis

PONE-D-23-11384R2

Dear Dr. Abebe,

We’re pleased to inform you that your manuscript has been judged scientifically suitable for publication and will be formally accepted for publication once it meets all outstanding technical requirements.

Kind regards,

Frank T. Spradley

Academic Editor

PLOS ONE

---

## [Editor Report · Acceptance letter]

8 Nov 2023

PONE-D-23-11384R2 

Prevalence and factors associated with early resumption of sexual intercourse among postpartum women: Systematic Review and Meta-Analysis 

Dear Dr. Abebe Gelaw:

I'm pleased to inform you that your manuscript has been deemed suitable for publication in PLOS ONE. Congratulations! Your manuscript is now with our production department. 

Kind regards, 

on behalf of

Dr. Frank T. Spradley 

Academic Editor

PLOS ONE